# Transcutaneous Auricular Vagal Nerve Stimulation in Healthy Non-Sedated Horses: A Feasibility Study

**DOI:** 10.3390/vetsci11060241

**Published:** 2024-05-28

**Authors:** Valentina Vitale, Francesca Bindi, Ana Velloso Alvarez, María de la Cuesta-Torrado, Giulia Sala, Micaela Sgorbini

**Affiliations:** 1Department of Animal Medicine and Surgery, Institute of Biomedical Sciences, Cardenal Herrera-CEU University, CEU Universities, 46115 Alfara del Patriarca, Spain; ana.vellosoalvarez@uchceu.es (A.V.A.); maria.de2@uchceu.es (M.d.l.C.-T.); 2Department of Veterinary Sciences, University of Pisa, Viale delle Piagge 2, 56122 Pisa, Italy; francesca.bindi@phd.unipi.it (F.B.); giulia.sala@unipi.it (G.S.); micaela.sgorbini@unipi.it (M.S.)

**Keywords:** transcutaneous electrical nerve stimulation, parasympathetic activity, equine, post-operative ileus, auricular vagal branch

## Abstract

**Simple Summary:**

Transcutaneous auricular vagal nerve stimulation is a non-invasive technique used in human medicine to prevent or treat post-operative ileus. The vagus nerve plays a crucial role in increasing intestinal motility and reducing the inflammatory response in sepsis models; thus, its stimulation constitutes an attractive strategy for modulating intestinal motility and immune response after abdominal surgery in both humans and horses. The technique has never been described in the equine species before. This study describes transcutaneous auricular vagal nerve stimulation in healthy non-sedated horses, assessing its feasibility and effect on heart rate variability (HRV). HRV was analysed as an indirect method to identify parasympathetic stimulation of the treatment. Auricular vagal nerve stimulation is an economical procedure that is easy to perform. However, it was poorly tolerated in awake horses. In the mares that tolerated the procedure, its effect on the autonomic system confirmed the potential stimulation of vagal activity, which can be of interest in preventing post-operative ileus in horses with inflammation or after exploratory laparotomy.

**Abstract:**

This study aimed to evaluate the feasibility of transcutaneous auricular vagal nerve stimulation (tAVNS) in healthy horses and its effect on heart rate variability (HRV). The study comprised three phases: the selection of mares, their acclimatization to the tAVNS, and the stimulation phase. Stimulation was performed with two electrodes positioned on the right pinna. The settings were 0.5 mA, 250 μs, and 25 Hz for pulse amplitude, pulse width, and pulse frequency, respectively. HRV was analysed before (B1), during (T), and after (B2) the tAVNS. From the 44 mares initially included, only 7 completed the three phases. In these mares, the heart rate (HR) was significantly lower, and frequency domain parameters showed an increased parasympathetic tone in B2 compared with B1. However, in 3/7 mares, the HR was significantly higher during T compared with B1 and B2, compatible with a decreased parasympathetic tone, while in 4/7 mares, the HR was significantly lower and the parasympathetic nervous system index was significantly higher during T and B2 compared with B1. The tAVNS is an economical and easy procedure to perform and has the potential to stimulate vagal activity; however, it was poorly tolerated in the mares included in this study.

## 1. Introduction

The term post-operative ileus (POI) describes the cessation or reduction in gastrointestinal (GI) transit following surgical stress [1]. The development of POI has been attributed to several causes and mechanisms. These may include anaesthetic agents, opioids, intravenous fluids, electrolyte imbalances, disruption to GI hormones and neuropeptides, disruption of neural continuity, autonomic dysfunction, and inflammatory cell activation [2]. However, it is currently accepted that the pathogenesis of POI involves two phases: an initial neurogenic phase resulting in immediate post-operative impairment of bowel motility and a subsequent inflammatory phase lasting for several days [1]. Despite representing distinct phases in POI progression, it is believed that the neuronal and immune systems exert a bi-directional interaction that might potentially help to reverse the ileus. During abdominal surgery, the surgical incision, peritoneal breach, and intestinal manipulation act as nociceptive stimuli that activate a sympathetic efferent response that results in reduced gastrointestinal motility [3]. The neuronal inhibition of GI motility is a self-limiting event, with normalisation of function upon cessation of nociceptor and mechanoreceptor stimulation, while the subsequent inflammatory response results in a significantly more prolonged negative effect on intestinal motility [1].

POI has become increasingly important and very challenging to manage in both veterinary and human patients [4]. According to recent surveys among European and American colleges of veterinary specialization, the perceived incidence of POI following emergency laparotomy in horses ranged between 0% and 20% [5,6]. The development of POI significantly impacts hospitalization time, treatment costs, and post-operative morbidity and mortality [1]. Despite the multifactorial nature of its pathology, a universal approach to POI management is lacking [5]. Current preventive and management measures in equine medicine, mainly relying on supportive care such as anti-inflammatories, prokinetic and antimicrobial drugs, and intravenous fluid therapy, are unfortunately often ineffective [4,5,6].

Research in human medicine has shown that early feeding can trigger vagus nerve stimulation which, in turn, ameliorates the local and systemic inflammatory response by binding acetylcholine (Ach) to α7 nicotinic Ach receptors expressed on macrophages. This results in the downregulation of tissue macrophage reactivity and cytokine release [7]. Therefore, sham feeding in the form of gum chewing in humans and bit chewing in horses has been employed to treat or prevent POI [4,7,8,9].

Numerous human studies support the theory that vagal tone can be influenced by auricular acupuncture [10,11,12,13]. Transcutaneous auricular vagal nerve stimulation (tAVNS) is a non-invasive technique involving the stimulation of the auricular branch of the vagus nerve (ABVN), which leads to reduced systemic inflammation [14]. As evidence that its effect on intestinal motility has been known for many years, the ABVN is sometimes called Alderman’s nerve, a reference to the centuries-old Aldermen of the city of London, who, during ceremonial banquets, encouraged attendees to place a napkin moistened with rosewater behind their ears in the belief this would aid digestion [10].

Protocols of tAVNS in humans have been performed for 10–30 min with pulse amplitude between 0.5 and 10 mA, pulse width between 250 and 500 μs, and frequency of 25 Hz. This method has been approved for the management of several clinical disorders, including epilepsy, depression, migraine, and tinnitus [10]. Recently, tAVNS has been introduced in both scientific research and clinical settings for the prevention and treatment of POI in humans [15].

To the best of the authors’ knowledge, there are no reports on the use of tAVNS in horses, nor a clear description of the anatomic location of the ABVN in this species. This research aimed to evaluate the feasibility of tAVNS in healthy non-sedated horses, with a final aim to assess the of tAVNS effect on the autonomic nervous system. To evaluate vagus stimulation, heart rate variability (HRV) was analysed prior to, during, and after tAVNS stimulation.

## 2. Materials and Methods

This study took place in August 2023 after receiving approval from the Research Ethical Committee of the University of Pisa (n. 8/23) on 27 February 2023.

Forty-four healthy mares (37 Standardbreds and 7 Thoroughbreds), owned by the University of Pisa, were initially included in this study. The mares varied in age (range 4–27 years) and body weight (range 522–568 kg) and were housed in large sand paddocks (20 × 30 m) with shelters, each allocated 5–8 mares. All the mares spent some hours or a few days in 4 × 4 m stables for clinical procedures, students’ practice, or investigations. Regardless of the location, all of them received the same diet with *ad libitum* grass hay and vitamin/mineral supplements. Health at the time of the study was assessed by administering a complete physical examination and blood analysis, including haematology and a biochemistry panel.

The study comprised three phases: (1) selection of mares tolerant to auricular manipulation, (2) acclimatization of the mares that tolerated the ear manipulation of tAVNS, and (3) stimulation phase of the mares that tolerated the tAVNS. This last phase lasted a total of 90 min, during which the HRV was recorded.

In the first phase, mares were led to a familiar stable with free access to water and grass hay. Inside the stable, the operators tried to gently touch both ears of the mares to test the tolerability for the placement of adhesive electrodes to perform the tAVNS. The mares who showed intolerance to this manipulation (jumps, head shaking, circling, or refusing contact with the operator) were immediately excluded from the study. Those mares who tolerated the ear manipulation with no avoidance behaviour proceeded to the second phase, where two adhesive electrodes were placed on the external and internal sides of the right pinna. This location was considered the most feasible for the adhesive patch used. Furthermore, to maintain the electrodes in position and avoid scratching, both ears were padded with cotton balls and an ear bonnet was placed on the head (Figure 1). The electrodes were then connected to the Rehabmedic EV906 device, which was placed on a neck collar (Figure 2). The device was set on normal transcutaneous electrical nerve stimulation (TENS) mode, with a pulse amplitude of 0.5 mA, pulse width of 250 μs, and pulse frequency of 25 Hz. Five minutes of stimulation was performed to assess if the mares tolerated the stimulation. During this period, the operator remained near the mare to check her tolerance to the procedure. In case of any sign of discomfort, the device was immediately switched off, the adhesive patch removed, and the mare excluded from the rest of the study. Only the mares who remained calm during the whole acclimatization period proceeded to the last phase, where they were left in the same stable with the TENS device in place but switched off.

For the third phase, the mares were equipped with a polar belt placed on the thorax behind the elbow, with ultrasound gel applied to the electrodes. The watch Polar Vantage V2 was connected by Bluetooth to the belt and attached to the head collar (Figure 2). 

This last phase was, in turn, divided into three steps (Figure 3): the polar watch for heart rate (HR) recording was started at T0 and the mare was left alone in the stable for 30 min. This constituted the control phase, with the TENS device in place but switched off (B1). After 30 min (T30), an operator quietly entered the stable to start 30 min of stimulation with the TENS device already set with the previously mentioned parameters (T). The device automatically switched off after 30 min (T60) and the mare was left alone for another 30 min (T90) for a further control phase with the TENS device still in place but without tAVNS (B2). Subsequently, both the polar belt and the TENS device were removed, and the mare was returned to her habitual paddock without further activity. During the whole 90-min period, an operator observed their behaviour from the outside of the stable and was ready to interrupt the procedure in case the subjects showed any sign of discomfort (vocalization, circling, jumps, etc.). The mares were free to move in the stall and had free access to water but not to food.

The HR recordings were extracted from the watch using the Polar dedicated software (Polar FlowSync version 4, Polar Electro). Heart rate variability (HRV) was analysed with the KubiosHRVStandard software (version 3.5.0, 2021 Kubios Oy) at intervals of 5 min in each 30-min session, according to previous studies [16]. The following parameters of the HRV analysis were selected: HR, standard deviation of R–R intervals (SDRR), root mean square of successive differences (RMSSD), standard deviations of the Poincaré plot (SD1 and SD2), high-frequency and low-frequency peaks in normalized units (HF and LF), LF/HF ratio, and the Baevsky’s stress index (STi). As no single HRV index provides an accurate description of the activation of each autonomic system [17], parameters derived from a combination of variables were also used. In particular, the sympathetic nervous system index (SNSi) is computed from mean HR, Baevsky’s stress index, and length of the distribution of Poincaré plots after non-linear analysis, while the parasympathetic nervous system index (PNSi) is computed from mean interbeat intervals, RMSSD in time-domain analysis, and the width of the distribution of Poincaré plots [18].

### Statistical Analysis

Statistical analysis was carried out using IBM SPSS v. 28.0 (IBM Corporation, Armonk, NY, USA). Descriptive statistics were performed, and continuous variables were expressed as the median, 25th, and 75th percentiles, as the data were not normally distributed (Shapiro–Wilk test). Differences between B1, T, and B2 were assessed using Generalized Linear Mixed Models (GLMM) for repeated measures. For each HRV parameter, a separate GLMM was conducted, with the groups as the fixed effect. The GLMM was applied to all subjects during phase 3.

Significance for all tests was set at a *p*-value < 0.05. 

## 3. Results

Only a total of 22/44 mares tolerated the manipulation of their ears and passed to the second phase. Among these 22 mares, 12 did not accept the tAVNS, thus only a total of 10 mares reached the third phase of the study. However, three additional subjects were excluded during the HRV recordings because they showed signs of discomfort and tachycardia (>80 bpm). Finally, HRV analysis was performed in only 7/44 mares. These were 5 Standardbreds and 2 Thoroughbreds aged between 11 and 19 years old. 

In these seven mares, HR was significantly lower in B2 compared with B1 (*p* = 0.012), while no difference was detected between B2 and T. SDRR and SD2 were decreased (*p* = 0.009 and 0.001, respectively) in B2 compared with B1 and T. HF was increased (*p* = 0.004), while LF and LF/HF were decreased (*p* = 0.004 and 0.014, respectively) in B2 compared with B1 and T. Furthermore, PNSi was increased (*p* = 0.009), while SNSi was decreased (*p* = 0.004) in B2 compared with B1, but no difference was detected between B2 and T. RMSSD, SD1, and STi did not show any differences across the three sessions. Complete results expressed as median and 25th–75th percentiles are reported in Table 1.

Analysing the HRV variability of the single subjects, we observed two different trends during the T session: 3/7 mares showed an increased HR, while 4/7 showed a decreased HR. Because of this unexpected difference, statistical analysis was repeated using a split GLMM, categorizing horses into those with an increase in HR (group A) and those with a decrease in HR (group B) during T. Subsequently, the GLMM for each heart rate variability parameter was recalculated for these two groups.

In group A, the HR was significantly higher during the T session compared with B1 (*p* = 0.028) and B2 (*p* < 0.001) sessions. Furthermore, in these three subjects, SDRR, RMSSD, SD1, and SD2 were significantly lower in B2 compared with B1 and T (*p* < 0.05) and STi was significantly higher in B2 compared with B1 and T (*p* = 0.003). 

In group B, HR was significantly lower during T and B2 compared with B1 (*p* = 0.005). There was an increase in HF (*p* = 0.004) and a decrease in LF (*p* = 0.004) and LF/HF (*p* = 0.005) during B2 compared with B1 and T, while no difference was detected between B1 and T. PNSi was significantly higher (*p* = 0.004) and SNSi was significantly lower (*p* = 0.008) during B2 and T compared with B1 (Figure 4). No differences were detected for SDRR, RMSSD, SD1, SD2, and STi across the three sessions.

## 4. Discussion

This study describes, for the first time, the application of tAVNS in healthy non-sedated horses. We assessed the tolerance of the mares included in this research to the TENS device and evaluated its effect on the autonomic nervous system by analysing the HRV. 

Many horses are ear-shy, making it challenging to touch their ears. Ear-shyness can stem from various reasons, including physical or behavioural problems, although the investigation of these factors was beyond the scope of this study. Nevertheless, this constitutes a big issue with the feasibility of the tAVNS in awake horses, as in the current study, only 7/44 mares safely tolerated the procedure until the end of the study. 

The remaining 37/44 mares were excluded because they showed signs of discomfort during one of the three phases. Mental stress is associated with a decreased parasympathetic tone [19], thus, to continue with the research protocol was considered not only unfair to their welfare but also against the scope of the research.

A wider population of horses of different breeds and sexes should be tested to confirm that the tAVNS is a procedure poorly tolerated by awake subjects of this species. 

Regarding the position of the electrodes, there is a lack of anatomical studies in horses. In humans, there are several cadaveric studies and research performed with functional magnetic resonance imaging (fMRI) that indicate the inner tragus and the cymba concha as active sites for vagal modulation [10]. In general, in mammals, it is believed that the ABVN courses through the mastoid canaliculus and then between the internal jugular vein and the bony wall of the jugular foramen, ultimately reaching the brain stem [20]. In dogs, TENS has been applied on the right tragus to treat the initial phase of atrial fibrillation [21]. However, in that study, the tAVNS was performed with the use of plastic clips in dogs under general anaesthesia. The tragus in horses was not an adequate location for the adhesive patch that we used in the current study, as due to their size and shape, they would not have remained in place. In humans, the earlobe has been widely used as a sham location, as it was believed to be relatively free of vagal afferents, but subsequently, it was found that stimulation at this level had an effect and produced similar fMRI patterns to ABVN stimulation [10]. Therefore, it has been speculated that electrical stimulation at any point in the ear might activate vagal afferents due to the dissemination of electrical current [11,15]. Based on these conclusions, we believed that, in the absence of anatomical indications, any location on the ear could have the potential to stimulate vagal afferent fibres, and the external and internal surfaces of the pinna were the most adequate for the application of the adhesive patches.

The settings used in the TENS device were extrapolated from research conducted in humans [10]. The HRV analysis of the seven mares included in the third phase of the study highlights significant differences between B2 and B1. In particular, in B2, there was a decrease in SDRR and SD2 in the time domain, an increase in HF, and a decrease in LF and LF/HF in the frequency domain. SDRR and SD2 are both influenced by sympathetic and parasympathetic activity [22]. However, the increase in HF and the decrease in LF have been reported to be compatible only with increased vagal and decreased sympathetic activity [16]. The interpretation of these results could be that there was vagal stimulation and much more significant sympathetic withdrawal in B2, as shown by the differences in PNSi and SNSi. However, these indexes are computed from the HR and the other HRV parameters [18]; thus, they are subjected to the same bias of human interpretation. A careful look at the single results for each horse is what allowed us to identify two subgroups with different autonomic responses. While no differences in the behaviour were observed, an increase in HR and a decrease in the time domain HRV parameters during T were detected in group A. These changes are compatible with decreased parasympathetic and increased sympathetic tones [16], which is the opposite of what was expected to be obtained with the tAVNS. 

On the other hand, the increased parasympathetic tone observed in group B during T and B2 was supported by the significant increase in HF and decrease in LF and LF/HF. Thus, we believe that the differences observed in the whole group of seven mares were just the expression of the differences detected in these subgroups and caution should be taken in the interpretation of the autonomic function. As a result, the tAVNS actually enhanced vagal tone in only 4/7 mares in the current study.

Reported side effects of the tAVNS in people included local skin irritation, itching, tinnitus, headache, and nasopharyngitis [23]. While no skin irritation nor itching were observed in the mares subjected to the tAVNS in this study, it cannot be ruled out that some of them experienced tinnitus or headache, which could explain the increased HR and decreased HRV in group A during T. It is possible that, regardless of the vagal nerve stimulation, the electrical vibration associated with the TENS caused some kind of discomfort in these mares, even if this was not detected from their behaviour. A lower pulse set should be tested in future studies to assess the lower limit of vagal stimulation without side effects. 

At the same time, several reports of the effects of TENS on the autonomic nervous system in humans have yielded inconsistent results, mainly due to the variations in stimulation frequency, intensity, and location at which TENS was applied [24,25]. All these differences could activate vagal afferents to different degrees [10]. It is known that the vagus nerve contains A, B, and C fibres. Up to 70–80% of the vagal fibres are unmyelinated C fibres, requiring higher stimulation strength and lower frequency for activation [26,27]. While direct activation of C fibres typically induces bradycardia, TENS primarily stimulates A fibres, which in turn suppress the pain transmission mediated by C fibres [28]. In people, the number of A myelinated axons in the ABVN can vary widely between individuals, which could explain why tAVNS may not be effective in some patients [29]. The same observations could be valid for horses, which can explain the differences observed in the HRV between groups A and B. Thus, until specific anatomical studies are performed on this species, we cannot rule out that, as it occurs in humans, tAVNS is effective in activating the vagus nerve only in some subjects.

According to the results of the HRV, activation of the vagal nerve in the four mares in group B started during the 30 min of stimulation (T) and remained sustained for the subsequent 30-min period (B2). Although this could have been simply associated with progressive relaxation in the stable, this was not observed in the mares in group A. Moreover, a sustained effect of the tAVNS on the parasympathetic tone has already been reported in humans [10]. This observation may be of particular interest in the context of its potential application in the prevention of POI in horses, as a short duration of tAVNS could be associated with a prolonged increase in vagal tone. 

Given that the gastric vagal efferent shares the same nucleus with the cardiac vagal efferent, an increased vagal efferent outflow, as assessed by HRV analysis, would suggest enhanced gastric vagal efferent activity [15,30]. The vagus nerve has been repeatedly shown to play a crucial role in increasing the motility of the stomach and intestine by initiating excitatory stimuli within the smooth muscle cells [31]. Furthermore, the vagus nerve was also shown to reduce the inflammatory response in a sepsis model, introducing the concept of the cholinergic anti-inflammatory pathway [32]. The prokinetic effect combined with the anti-inflammatory properties of the vagus nerve constitutes an attractive strategy for modulating intestinal motility and immune responses after abdominal surgery in both humans and horses. As the current study does not support the use of tAVNS in awake horses due to their ear-shyness, further studies can evaluate its feasibility in horses under general anaesthesia due to acute gastrointestinal disease or in the immediate post-operative period, when the residual anaesthetic effect may render the horses less sensitive to electrical vibrations in the ears.

This study has several limitations. First of all, only a small number of animals were included in the third phase, thus the power of the study is limited and results should be confirmed in a larger population. Moreover, the behaviours of the animals included in this study were only assessed by the veterinarians performing the research but were not recorded for successive evaluation or blind assessment. Although the behavioural differences between the horses that tolerated the ear touch and electrical stimulation and those that did not were evident, it is possible that subtle differences were missed and could have explained the differences in HRV observed between groups A and B. Furthermore, we did not evaluate other possible effects of the vagal stimulation such as intestinal motility and faecal production. However, auscultation of the abdomen during the tAVNS was precluded because of the effect that human interactions can have on the HRV [16]. Evaluation of faeces production would have required prolonged stabling of the mares, which was not possible because it did not fit within the institutional organization.

## 5. Conclusions

In conclusion, the tAVNS is an economical and easy procedure to perform, although it was poorly tolerated in the horses included in this study. It is possible that the mares used in this study were particularly sensitive to manipulation of the ears, thus, additional studies in a different population of horses should be performed. Alternatively, a lower pulse amplitude and frequency could be tested. Furthermore, among the mares that tolerated the tAVNS, vagal tone was enhanced in only 4/7. Further studies are needed to confirm these results and to investigate the effects of tAVNS in horses with gastrointestinal disease before its use can be recommended for the prevention or treatment of POI in the equine species.

## Figures and Tables

**Figure 1 vetsci-11-00241-f001:**
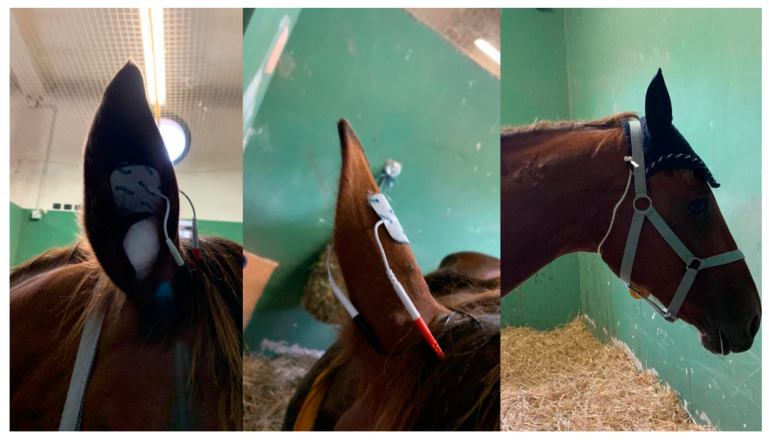
Position of the two adhesive electrodes on the external and internal sides of the right pinna. Ears were padded with cotton balls and an ear bonnet was placed on the head.

**Figure 2 vetsci-11-00241-f002:**
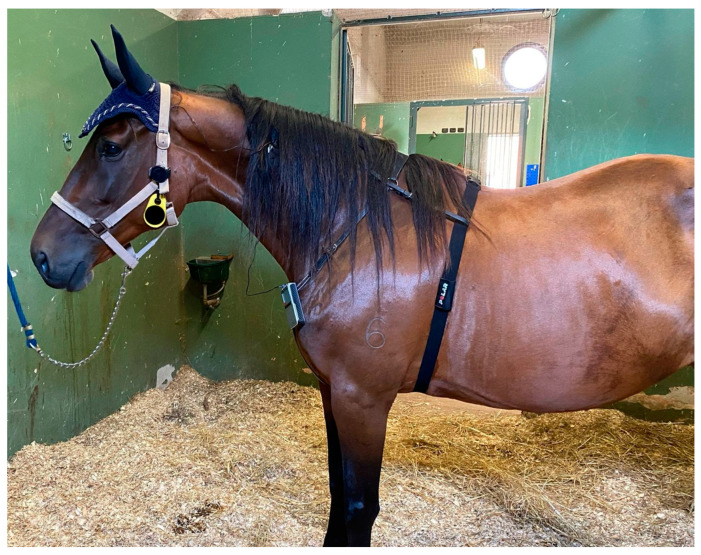
The TENS device was placed on a neck collar and the polar belt was placed on the thorax behind the elbow, with ultrasound gel applied to the electrodes. The watch Polar Vantage V2 was connected to the belt and attached to the head collar.

**Figure 3 vetsci-11-00241-f003:**
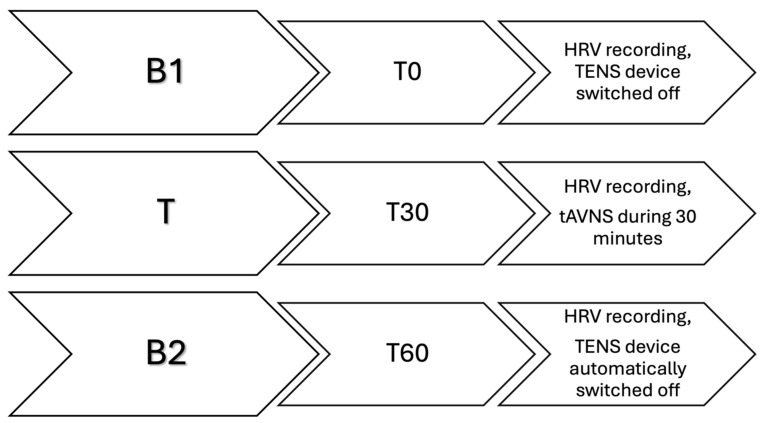
Graphical representation of the three steps of the third phase of the study. Each step lasted 30 min. B1 (from T0 to T30) constitutes the first basal recording of heart rate variability (HRV) with the TENS device in place but switched off. T (from T30 to T60) constitutes the treatment period, during which the HRV is recorded while the transcutaneous auricular vagal nerve stimulation takes place. B2 (from T60 to T90) constitutes the second basal recording of the HRV, with the TENS device automatically switched off.

**Figure 4 vetsci-11-00241-f004:**
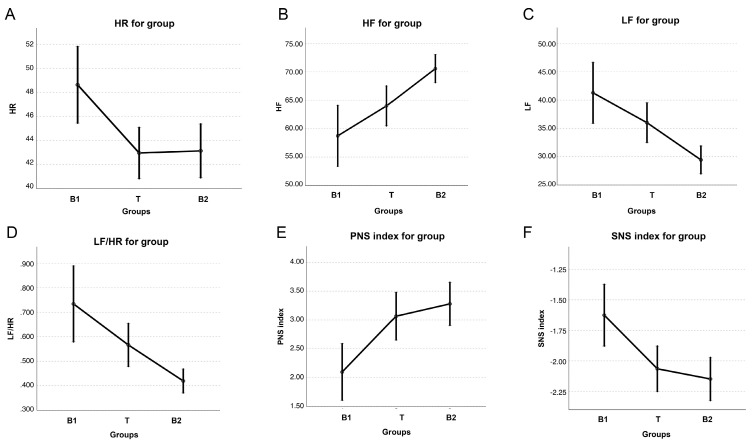
Graphical representation of the following parameters of heart rate variability: heart rate (HR) (**A**), high frequency (HF) (**B**), low frequency (LF) (**C**), LF/HF ratio (**D**), parasympathetic nervous system index (PNSi), and (**E**) sympathetic nervous system index (SNSi) (**F**), expressed as median (dots) and 25th–75th percentiles (bars) in 4/7 mares during the first basal recording (B1), TENS treatment (T), and second basal recording (B2).

**Table 1 vetsci-11-00241-t001:** HRV analysis results expressed as medians (25th–75th percentiles) of the 7/44 mares during the first basal recording (B1), the TENS treatment (T), and the second basal recording (B2). A different superscript letter (a or b) on the same row indicates a statistically significant difference. Legend—NS: not significant, HR: heart rate, SDRR: standard deviation of R–R intervals, RMSSD: root mean square of successive differences, SD1 and SD2: standard deviations of the Poincaré plot, HF (n.u.): high-frequency peaks in normalized units, LF (n.u.): low-frequency peaks in normalized units, PNSi: parasympathetic nervous system index, SNSi: sympathetic nervous system index, STi: Baevsky’s stress index.

	B1	T	B2	Statistical Significance
HR (bpm)	40.5 (38.7–53.0) ^a^	41.0 (38.0–43.0) ^a,b^	40.0 (35.0–43.0) ^b^	*p* = 0.012
SDRR (ms)	60.2 (52.7–88.8) ^a^	67.7 (54.7–83.8) ^a^	55.6 (44.7–67.6) ^b^	*p* = 0.009
RMSSD (ms)	68.5 (52.5–94.1)	72.5 (62.8–89.4)	71.4 (58.3–82.5)	NS
SD1 (ms)	48.5 (37.2–66.7)	51.4 (44.5–63.4)	50.6 (41.4–58.5)	*p* = 0.014
SD2 (ms)	73.6 (60.1–108.4) ^a^	78.4 (61.3–100.6) ^a^	61.0 (45.3–75.9) ^b^	NS
HF (n.u.)	61.1 (51.7–74.1) ^a^	63.6 (55.9–74.8) ^a^	71.4 (65.8–80.6) ^b^	*p* = 0.004
LF (n.u.)	38.9 (25.9–48.3) ^a^	36.4 (25.2–44.1) ^a^	28.6 (19.4–34.1) ^b^	*p* = 0.004
LF/HF	0.6 (0.3–0.9) ^a^	0.6 (0.3–0.8) ^a^	0.4 (0.2–0.5) ^b^	*p* = 0.014
PNSi	2.9 (2.3–3.7) ^a^	3.1 (2.6–4.0) ^a,b^	3.5 (2.9–4.3) ^b^	*p* = 0.009
SNSi	−2.0 (−2.4–−1.6) ^a^	−2.2 (−2.4–−1.9) ^a,b^	−2.3 (−2.5–−2.1) ^b^	*p* = 0.004
STi	5.8 (4.0–6.8)	5.3 (4.4–6.4)	6.1 (5.5–6.8)	NS

## Data Availability

The original contributions presented in the study are included in the article, further inquiries can be directed to the corresponding author.

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
