# Peer review of "Transcutaneous Auricular Vagal Nerve Stimulation in Healthy Non-Sedated Horses: A Feasibility Study"

_vetsci, 2024, doi:10.3390/vetsci11060241_

Round 1

Reviewer 1 Report

Comments and Suggestions for Authors

1.     Rewrite the abstract, the language needs to be polished. Additionally, there’s no need to specify the ‘background’, or ‘Results’ in the abstract. The authors should integrate them in one paragraph and use the language attractive enough for the readers.

2.     In Figure 3, the authors should denote the meaning of the bars? It is crucial to specify whether the bars represent standard deviation (SD) or standard error of the mean (SEM).

3.     In Table 1, please denote the meaning of groups B1, T, B2 in the legend. Furthermore, the meaning of the superscripts 'a' and 'b' should be elucidated within the legend to enhance reader comprehension.

4.     It is important to note that all the experiments were carried out according to the guidance set by the corresponding ethics committee. It is important to consider the welfare of experimental animals.

5.     The authors should include more discussions on the statement in the conclusion part “The tAVNS is a cheap and easy procedure to perform, although poorly tolerated in awake horses. However, it could have a potential application under general anesthesia.” I appreciate the authors pointing out the limitations of their current research however I believe it is more important to discuss how modern technology development could benefit their current research. To my knowledge, noninvasive wearable devices based on soft electrodes are under fast development in both academia and industry. Those devices function properly in awake human beings and animals. The authors should include more discussion about how those devices can help to improve their current research or what are the limitations to applying those new devices in their research.

Comments on the Quality of English Language

The abstract needs to be rewritten.

Author Response

Thank you for you comments.

Below the specific replies:

  1. Rewrite the abstract, the language needs to be polished. Additionally, there’s no need to specify the ‘background’, or ‘Results’ in the abstract. The authors should integrate them in one paragraph and use the language attractive enough for the readers.

This has been done. Subtitles have been removed and the text rewritten integrated as a single paragraph.

  1. In Figure 3, the authors should denote the meaning of the bars? It is crucial to specify whether the bars represent standard deviation (SD) or standard error of the mean (SEM).

This was already indicated, however, we now specified it better in this way:

"expressed as median (dot) and 25th-75th percentiles (bars)"

  1. In Table 1, please denote the meaning of groups B1, T, B2 in the legend. Furthermore, the meaning of the superscripts 'a' and 'b' should be elucidated within the legend to enhance reader comprehension.

The meaning of B1, T and B2 has been added and the use of a and b has been specified as it was wrongly indicated in some of the lines.

  1. It is important to note that all the experiments were carried out according to the guidance set by the corresponding ethics committee. It is important to consider the welfare of experimental animals.

Yes, the experiment was carried out after receiving approval from the Research Ethical Committee of the University of Pisa (n. 8/23) on 27/02/2023.

  1. The authors should include more discussions on the statement in the conclusion part “The tAVNS is a cheap and easy procedure to perform, although poorly tolerated in awake horses. However, it could have a potential application under general anesthesia.” I appreciate the authors pointing out the limitations of their current research however I believe it is more important to discuss how modern technology development could benefit their current research. To my knowledge, noninvasive wearable devices based on soft electrodes are under fast development in both academia and industry. Those devices function properly in awake human beings and animals. The authors should include more discussion about how those devices can help to improve their current research or what are the limitations to applying those new devices in their research.

We include more discussion the potential application of the device (lines 357-371) and its poor tolerability in the mares of this study (lines 325-348)

Reviewer 2 Report

Comments and Suggestions for Authors

The presented manuscript stands out for being an original and interesting topic.

The main deficiency of the work is in its statistical design and its execution. The data must be fully evaluated and the analysis must be described correctly to clarify some inconsistencies. 

It is not clear why the authors make the division between those who raise HR and those who do not, and whether it is statistically correct. The discussion is mostly based on that. There are inconsistencies in the number of animals, and in results presented, which are then discussed in this way.

The approach to horses with post-colic ileus is not clear.

The use of phases to explain the investigation could be more careful. It is used to separate the preparation of the animals, and to prepare the registration stages. this is confusing.

L94 Influenza vaccination is incorrect. It is better not to mention it to avoid confusion.

Authors must detail the exclusion factors precisely. when they considered that animals were not docile enough?

Electrodes were very close... did the authors check the nerve stimulation?

L96 full blood work analysis... please detail

L167 7 or 10? 

L178 table1... Statistics from HR, SD1, SDRR are ok?

L186 are compatible with... this is discussion

L190 The use of the group of four mares is not described in m&m, and is not properly founded.

Author Response

Thank you for the comments. Below the replies.

The presented manuscript stands out for being an original and interesting topic.

The main deficiency of the work is in its statistical design and its execution. The data must be fully evaluated and the analysis must be described correctly to clarify some inconsistencies. 

We tried to clarify and discuss better all the inconsistencies of the statistic analysis, both in material and methods and in results.

It is not clear why the authors make the division between those who raise HR and those who do not, and whether it is statistically correct. The discussion is mostly based on that. There are inconsistencies in the number of animals, and in results presented, which are then discussed in this way.

This has been clarified (lines 255-270)

The approach to horses with post-colic ileus is not clear.

We discuss it better (lines 357-371)

The use of phases to explain the investigation could be more careful. It is used to separate the preparation of the animals, and to prepare the registration stages. this is confusing.

We explained it better in the M&M (lines 138-192) and we added a figure to explain the third phase of the study and avoid confusion.

L94 Influenza vaccination is incorrect. It is better not to mention it to avoid confusion.

This has been removed.

Authors must detail the exclusion factors precisely. when they considered that animals were not docile enough?

This has been better defined: "The mares who showed intolerance to this manipulation (jumps, head shaking, circling, or refusing the contact with the operator) were immediately excluded from the study" (lines 145-147)

Electrodes were very close... did the authors check the nerve stimulation?

This is similar to what has been done in humans and dogs (Zhu Y, Xu F, Lu D, Rong P, Cheng J, Li M, et al. Transcutaneous auricular vagal nerve stimulation improves functional dyspepsia by enhancing vagal efferent activity. Am J Physiol Gastrointest Liver Physiol. 2021;320(5):G700-g11. Epub 2021/02/25; Yu L, Scherlag BJ, Li S, Fan Y, Dyer J, Male S, et al. Low-level transcutaneous electrical stimulation of the auricular branch of the vagus nerve: a noninvasive approach to treat the initial phase of atrial fibrillation. Heart Rhythm. 2013;10(3):428-35).

As there are no anatomical studies on the vagal fibers location at this level in horses the current study aimed to check the nerve stimulation by evaluating the vagal activity with the heart rate variability.

L96 full blood work analysis... please detail

This has been detailed as follows: "blood analysis including haematology and a biochemistry panel"

L167 7 or 10? 

There were ten mares who proceeded to the third phase but three more have been excluded. This has been explained better (lines 228-234)

L178 table1... Statistics from HR, SD1, SDRR are ok?

The way the superscript letters were written was misleading. We corrected it as the only statistically difference was between B2 and B1 but not between B1 and T nor B2 and T. This has been corrected.

L186 are compatible with... this is discussion

This has been removed and added to the discussion (lines 318-324).

L190 The use of the group of four mares is not described in m&m, and is not properly founded.

This was not properly explained. It has been clarified in the results (lines 255-270) and discussed (lines 325-356).

Reviewer 3 Report

Comments and Suggestions for Authors

the authors evaluated transcutaneous vagal nerve stimulation in horse through the application of sensors over the ear of a cohort of unsedated  patients. they tried to evaluated the feasibility of the technique and the effects on HRV . The study is well described and the results are clear and exhausting.

I only have a couple of concerns about the statical validity of the study.

Since the authors first enrolled 44 horses, and ended with 7 completing the study, do they questioned about the power of the study with such a limited number of patients?

Moreover of these 7 mares 3 had a results opposite to the other 4. how could you explain it?

LIne 213: did you observed the behavior of the horses after stimulation? this could have highlighted some discomfort.

Author Response

Thank you for the comments. Below you can find the replies to each specific comment.

The authors evaluated transcutaneous vagal nerve stimulation in horse through the application of sensors over the ear of a cohort of unsedated  patients. they tried to evaluated the feasibility of the technique and the effects on HRV . The study is well described and the results are clear and exhausting.

I only have a couple of concerns about the statical validity of the study.

Since the authors first enrolled 44 horses, and ended with 7 completing the study, do they questioned about the power of the study with such a limited number of patients?

Yes, this a main limitation of the study, we commented it in the discussion (lines 372-375)

Moreover of these 7 mares 3 had a results opposite to the other 4. how could you explain it?

This was a point questioned also by the other reviewers thus we reorganized both the results and discussion and we tried to explain this difference (lines 325-356)

Line 213: did you observed the behavior of the horses after stimulation? this could have highlighted some discomfort.

We could not identify any sign of discomfort by the behaviour of these mares but, as it was not mentioned before, we included it in the discussion (lines 329-331 and 375-381)

Round 2

Reviewer 3 Report

Comments and Suggestions for Authors

The authors replied to the comments that I previously made. Compared to the first version  of the manuscript, they provided much more information about literature background, anatomical reason for the location of the sensors. But most of all they provided a much more consistent self analysis of the limitation of the study. I think that at the moment the manuscript can be published. Despite the several limitation and doubts about the results, they surely put the attention on a potential adjunctive way to treat or prevent POI.

Author Response

Many thanks for your comments and suggestions, we are happy to have addressed all your previous concerns.